# Primitive Cutaneous (P)erivascular (E)pithelioid (C)ell Tumour (PEComa): A New Case Report of a Rare Cutaneous Tumor

**DOI:** 10.3390/genes13071153

**Published:** 2022-06-26

**Authors:** Gerardo Cazzato, Anna Colagrande, Lucia Lospalluti, Lucia Pacello, Teresa Lettini, Francesca Arezzo, Vera Loizzi, Carmelo Lupo, Nadia Casatta, Gennaro Cormio, Eugenio Maiorano, Giuseppe Ingravallo, Leonardo Resta

**Affiliations:** 1Section of Molecular Pathology, Department of Emergency and Organ Transplantation (DETO), University of Bari “Aldo Moro”, 70124 Bari, Italy; anna.colagrande@policlinico.ba.it (A.C.); teresa.lettini@uniba.it (T.L.); eugenio.maiorano@uniba.it (E.M.); giuseppe.ingravallo@uniba.it (G.I.); leonardo.resta@uniba.it (L.R.); 2Section of Dermatology and Venereology, Department of Biomedical Sciences and Human Oncology (DIMO), University of Bari “Aldo Moro”, 70124 Bari, Italy; l.lospalluti@policlinico.ba.it (L.L.); lucia.pacello@policlinico.ba.it (L.P.); 3Section of Gynecology and Obstetrics, Department of Biomedical Sciences and Human Oncology (DIMO), University of Bari “Aldo Moro”, 70124 Bari, Italy; francesca.arezzo@uniba.it (F.A.); vera.loizzi@uniba.it (V.L.); 4Innovation Department, Diapath S.P.A., Via Savoldini n.71, 24057 Martinengo, Italy; carmelo.lupo@diapath.com (C.L.); nadia.casatta@diapath.com (N.C.); 5Gynecologic Oncology Unit, IRCCS Istituto Tumori “Giovanni Paolo II”, 70124 Bari, Italy; gennaro.cormio@uniba.it; 6Department of Interdisciplinary Medicine (DIM), University of Bari “Aldo Moro”, 70124 Bari, Italy

**Keywords:** PEComa, skin, primitive, cutaneous, sugar tumour, mesenchymal, clear cell

## Abstract

Perivascular epithelioid cell tumours (PEComas) are a growing family of tumours composed of histologically and immunohistochemically distinctive perivascular epithelioid cells. Cutaneous primitive PEComas (cPEComas) are very rare, with 65 cases described in the English literature, and occur as a painless lesion predominantly in female patients, with a wide age range. We present a new case of cPEComa found on the left thigh of a 53-year-old patient with histopathological, immunohistochemical, and molecular information. The lesion was positive for HMB-45 and focal for smooth muscle actin and desmin but negative for melan-A, S-100 protein, CD31, and CD34. Next generation sequencing (NGS) analysis demonstrated the presence of genomic aberration for baculoviral IAP repeats containing BIRC3 splice site 1622-27_1631del37. Although there are little molecular data regarding this entity, our case adds to this knowledge, considering the importance of detecting genomic aberrations in the context of specific therapies such as mTOR inhibitors.

## 1. Introduction

PEComas are mesenchymal tumours composed of perivascular epithelioid cells (PECs), a distinct group of cells that is often associated with the blood vessel wall and has the peculiar characteristic of simultaneously expressing melanocytic and smooth muscle immunohistochemical markers [1]. Actually, entities such as “angiomyolipoma”, “epithelioid angiomyolipoma” and “lymphangioleiomyomatosis” are included in this group, but, from 2020, with the release of the new classification of the World Health Organization, the use of old definitions such as “clear cell myomelanocytic tumour” or “sugar tumour of the lung” is not recommended [1]. PEComas are rare but more frequent in females than males, with an M:F ratio of 0.2:1, a wide age range, and a peak in young to middle-aged adults (mean age: 45 years) [2,3,4,5]. Primitive cutaneous PEComas (cPEComas) are rare, first described by Crowson et al. [6], and occur as a painless lesion predominantly in female patients, with a wide age range [7]. The majority of cPEComas arise on the extremities (lower > upper) and, less frequently, on the trunk [7,8]. In a very recent review [9], 65 cases of cPEComa were isolated from an in-depth analysis. In this paper, we present a new case of cPEComa found on the left thigh of a 53-year-old patient; we report its histopathological, immunohistochemical and molecular biology characteristics, and, finally, we compare our case with the cases previously reported in the literature.

## 2. Materials and Methods

A 53-year-old woman was presented to the dermatology and venereology U.O.C. due to the presence of a nodular neoformation, about 0.8 cm, slightly painful to the touch, in correspondence with the left thigh, which the patient reported to have been present for about 3 months. The patient did not report a significant previous medical history except for multiple sclerosis. There were no palpable axillary lymph nodes. The dermatologist hypothesised a possible leiomyoma, dermatofibroma, or transforming nevus. The patient and the dermatologist, therefore, decided to remove the lesion in question. An excisional biopsy was performed, and the sample, fixed in 10% neutral buffered formaldehyde, was sent to the U.O.C. of pathological anatomy. After sampling and processing, the sample was dipped in paraffin, and 5 µn thick sections were cut and stained with hematoxylin/eosin. In addition, other sections were prepared for immunostaining with antibodies to melan-A (MART1), human melanoma black antigen-45 (HMB-45), CD68 (PGM-1), S-100 protein, CD31, CD34, CD10, ki-67+, muscle-smooth actin, actin HHF-35, CK-pool and CK-7. The characteristics of the antibodies used are summarised in Table 1.

Finally, next generation sequencing (NGS) analysis was performed with NextSeq 550Dx (Illumina, San Diego, CA, USA). Data were analysed using NextSeq 550 RUO software (Illumina, San Diego, CA, USA).

## 3. Results

Histologically, the lesion was relatively sharply demarked from the surrounding tissue, extending into the subcutaneous tissue (Figure 1). The tumour consisted of epithelioid cells containing eosinophilic or clear granular cytoplasm, with round and vesicular nuclei and small nucleoli (Figure 2A,B). The tumour cells grew around thick-walled capillaries and sometimes infiltrated the subcutis (Figure 2C). There was no necrosis and no mitotic figures (Figure 2D). From an immunophenotypic point of view, the cells constituting the lesion expressed at least one melanocytic marker (in our case, constituted by HMB-45) (Figure 3A), while they were negative for melan-A (MART1) and S-100 protein (not shown). On the other hand, the lesion was focally positive for muscle-smooth actin (Figure 3B) and desmin (Figure 3D) but was negative for actin HHF-35 (not shown). There was an accompanying positivity of histiocytes (CD68+) (Figure 3C). CD10 was partially positive. CD31 and CD34 showed the vascular texture around which the lesion in question was present. Correlation of the morphological and immunophenotypic findings, particularly the co-expression of melanocytic and muscle markers, established a diagnosis of cPEComa. A second step consisted in defining this lesion in the spectrum of benignity or malignancy. In this regard, the scheme of Folpe et al. was used [3,4,5,6,7,8,9,10,11].

The cPEComa had only one characteristic suggestive of possible malignancy, the infiltrative growth pattern, but other parameters such as necrosis, tumour size greater than 5 cm and vascular invasion were absent. Furthermore, the patient was referred to a medical oncologist for systemic evaluation with computerised tomography (CT) and positron emission tomography (PET) of the thorax, abdomen and pelvis. All the scans were negative for metastases. Therefore, it ended in the diagnosis of benign cutaneous primitive PEComa.

After diagnosis, it was decided to send part of the sample for biomolecular investigations using next generation sequencing (NGS), which revealed genomic aberration for baculoviral IAP repeats containing BIRC3 splice site 1622-27_1631del37, the same aberration detected by Cohen et al. recently. In contrast, there were no mutations and/or aberrations related to tuberous sclerosis complex 1 (TSC1).

The patient underwent a follow-up one month after the diagnosis, and the picture remained unchanged.

## 4. Discussion

PEComatous tumours are thought to arise from perivascular epithelioid cells (PECs) [1,12]. For many years, clear cell tumours have been called sugar tumours in the lung and myomelanocytomas at other sites [12,13]. On the other hand, cPEComa is a rare tumour, first described in 2003 [6] and later confirmed by other reports [7,8,9,10,11,14,15,16,17]; it is not yet clear what the real histogenesis of this entity is, as the benign cell counterpart of origin of this neoplasm is not known [10]. The concept of PEComa was coined in a unifying sense for a series of different entities that has been named in the most diverse ways [3].

As far as cPEComa is concerned, the knowledge of its existence is of fundamental importance, especially in the context of the differential diagnosis of other neoplastic lesions that it can be confused with. For example, it is essential to conduct a correct differential diagnosis between cPEComa and neoplastic lesions with clear cell modifications, such as granular cell tumours and clear cell variants of granular cell tumours [18,19]. These tumours are non-encapsulated and composed of irregularly arranged sheets of large polyhedral cells with a small central hyperchromatic nucleus and abundant fine-to-coarse granular eosinophilic cytoplasm [18,19]. Furthermore, in addition to morphological differences compared to cPEComa, also from an immunophenotypic point of view, granule cell tumours are usually positive for S-100 protein, neuron-specific enolase (NSE) and melanoma-antigen associated with NKI/C-3, but there is co-expression with muscle markers [19].

Equally important is the differential diagnosis with lesions such as the granular cell variant of leiomyoma [20], leiomyosarcoma [15,21], uterine leiomyosarcoma itself [22], epithelioid fibrous histiocytoma (EFH) [23,24] (in the latter case, the strong positivity for CD68 PGM-1 and CD163 will be rather clear for an EFH), and peripheral nerve sheath tumours [24].

Differential diagnoses with entities such as malignant melanomas (MMs) and clear cell and balloon cell melanomas are equally important, considering the extensive polymorphic picture with which MMs can present [8,10,15,19,25,26]. In this case, a correct immunophenotypic investigation may prove sufficient since S-100 protein and Sry-related HMg-box gene 10 (SOX-10) will be positive in one case and not the other. In the setting of differential diagnosis with metastatic neoplasms, an important step is to exclude metastasis of renal cell carcinoma, a clear cell variant, a pitfall that can create diagnostic problems. In particular, considering that various authors have described cPEComa positivity to CD10 (a marker widely used in the diagnostics of renal neoplastic pathology), it is necessary to conduct further investigations with markers such as PAX-8 [19].

Equally important is the differential diagnosis with clear cell sarcoma, a neoplasm consisting of polygonal-to-fusiform cells that grow in nests or fascicles and that, similar to cPEComa, express HMB-45 and melan-A. However, it also typically demonstrates positive staining for both S-100 protein and SOX-10. Additionally, in contrast to cPEComa, clear cell sarcoma is negative for smooth muscle markers. In addition, if necessary, documentation of EWS–ATF1 gene fusion can definitely establish the diagnosis of clear cell sarcoma [8,27].

Another entity with which to make a differential diagnosis is represented by alveolar soft tissue sarcomas (ASTSs), a rare malignant mesenchymal neoplasm, frequently composed of large, polygonal cells with abundant eosinophilic cytoplasm, a nested or pseudoalveolar growth pattern, and PASD+ intracytoplasmic rhomboid- or rod-shaped crystals [28]. ASTS predominantly affects the deep soft tissues of the extremities (thigh and buttock) in young adults and the head and neck region (tongue and orbit) in children, and it is characterised by ASPSCR1–TFE3 gene fusion [28]. Immunohistochemically, ASTS demonstrates nuclear immunoreactivity for TFE3 [29] and cathepsin K, but it is negative for melanocytic and muscle markers [28].

cPEComa must be differentiated from skin metastasis by gastrointestinal stromal tumours (GISTs): about 10% of GISTs can metastasise to the skin [30,31]. In the metastatic setting, however, GISTs retain their characteristic immunophenotypic spectrum (positive for DOG-1 and CD117) and also other possible markers, including CD34 (70%), SMA (30-40%), S-100 protein (5%) and desmin (1–2%) [31].

ALK-rearranged cutaneous soft tissue tumours are neoplasms characterised by the presence of myxoid spindle cell whorls and cords and the co-expression of ALK, CD34, and, frequently, S100 protein and the so-called “superficial ALK-rearranged myxoid spindle cell neoplasm”. There are hardly any difficulties in the differential diagnosis of this entity due to different immunophenotypic characteristics compared to cPEComa [32].

From a molecular point of view, a paper by Akummalla et al. [33] reported a set of different genomic aberrations in a total of 31 patients analysed, reporting an average of 3.2 alterations/patient. The most common molecular alterations consisted of: the TSC2 gene, 32.3% of cases; TSC1, 9.6% of cases; TFE3 fusions in 16.1% of cases; and folliculin (FLCN) in 6.4% of cases. It is interesting to note that in our case, none of these aberrations were found, but an alteration of BIRC3 splice site 1622-27_1631 del37 (a gene encoding a protein with the function of inhibiting the mechanism of apoptosis by binding to the tumour necrosis receptor-associated factor) was found, similar to what was recently demonstrated by Cohen et al., who, unlike our case, presented a case of malignant cPEComa. This apparent disarray does not allow us to draw conclusions, but we report the data anyway as the data suggest we should still be cautious before stating the exact character of the presence of this mutation (pathogenic, malignant or benign?).

Table 2 summarises some examples of cPEComas presented in this work, with clinical, morphological and molecular information.

## 5. Conclusions

Cutaneous primitive PEComas represent a very rare entity of the already rare systemic PEComas. To date, to our knowledge, only 65 cases of CPEC have been reported, but they are certainly “legitimate”, to which the present case is added. Although there are little molecular data regarding this entity, our case adds to this knowledge, considering the importance of detecting genomic aberrations in the context of specific therapies such as mTOR inhibitors [34]. Further and future studies with larger case series will be needed to obtain new information and increase our knowledge of this rare entity.

## Figures and Tables

**Figure 1 genes-13-01153-f001:**
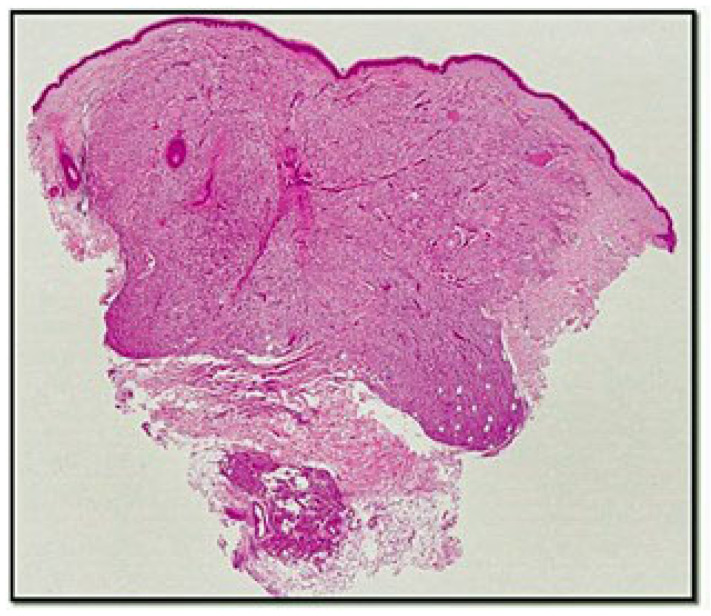
Histological micrograph showing a dermal, clear cell lesion infiltrating the subcutis at one point (hematoxylin–eosin, original magnification: 4×). Black arrow indicates the adipocytic lobules of the subcutis buried by the infiltration of the lesion.

**Figure 2 genes-13-01153-f002:**
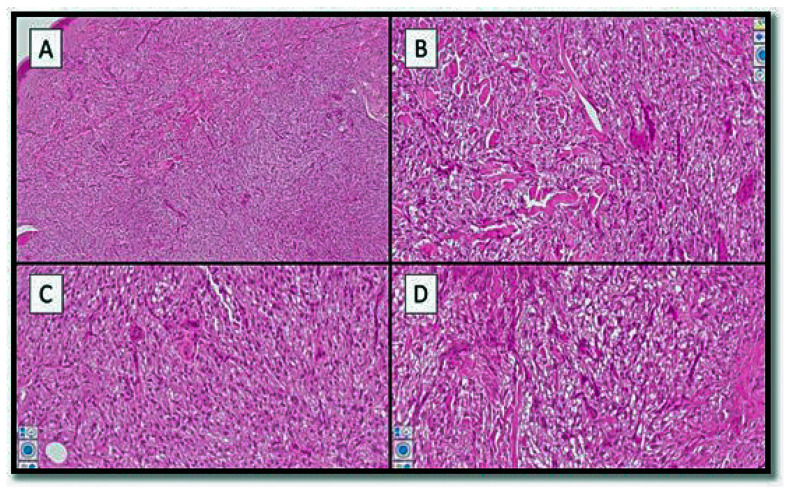
(**A**,**B**) The tumour was dermal-based, with a clear cell appearance and a chatacteristic trabecular pattern in which the tumour cells were arranged around numerous vessels (**C**,**D**) (hematoxylin–eosin, original magnification: 5×, 10×, 20×, and 40× respectively).

**Figure 3 genes-13-01153-f003:**
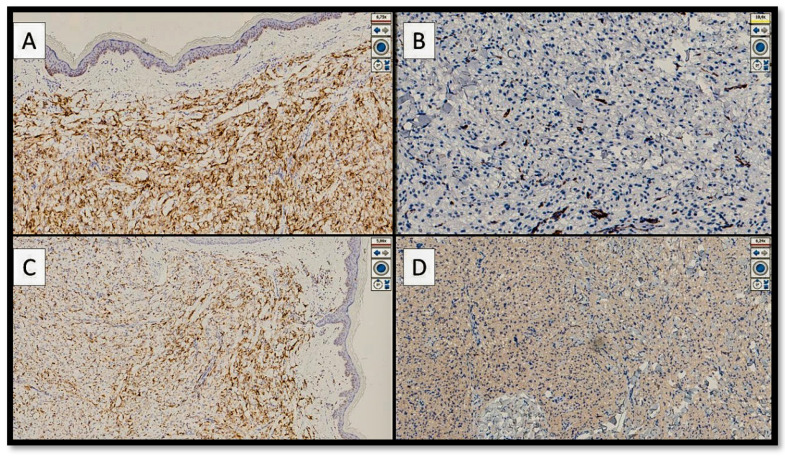
(**A**) Immunostaining photomicrograph with anti-HMB-45 antibody. Note the widespread and intense positivity of the clear epithelioid cells of the cPEComa (immunostaining for HMB-45, original magnification: 10×). (**B**) Immunostaining micrograph with anti-smooth muscle actin antibody. Note the almost complete negativity, with partial positivity of some elements (red arrows) (original magnification: 20×). (**C**) Photomicrograph shows positivity of CD68 (PGM-1) histiocytes accompanying the lesion (original magnification: 10×). (**D**) Photomicrograph shows almost complete negativity for desmin, with the focal and discontinuous positivity of some elements (original magnification: 20×).

**Table 1 genes-13-01153-t001:** Summary of antibodies used in this study.

Type of Antibodies Used	Fibrohistiocytic	Vascular	Neural/Melanocytic	Epithelial
	Anti-CD68 (PGM-1; Dako, 1:800)	Anti-CD34 (QBEND/10, Dako, 1:100)	Anti-S-100 protein (Polyclonal, Dako, 1:500)	Anti-EMA (E29, Dako, 1:200)
Immunohistochemical Markers (Clone, manufacturing company, dilution)	Anti-CD163 (Polyclonal, Thermofisher 1:50)	Anti-CD31 (JC70A, Dako, 1:40)	Anti-SOX-10 (Polyclonal Thermofisher, 1:2000)	Anti-CK-Pool (M3515, Dako, 1:150)
	Anti-Desmin (PA5-16705, Thermofisher, 1:200)		Anti-Melan-A (M2-7C10 + M2-9E3, ThermoFisher, 1:100) Anti-HMB-45 (ThermoFisher, 1:100)	Anti-CK 7 (Clone OV-TL 12/30, Dako, 1:100)

**Table 2 genes-13-01153-t002:** Some examples of cPEComas reported in the literature.

Author(s)	Number of Patients	Age	Gender	Localisation	Clinical Appearance	Histological Diagnosis	Outcome	Molecular Biology
Mentzel et al.	7	30-66	F	5 lower leg 1 popliteal fossa 1 forearm	not reported	Benign PEComa	indolent	not performed
Liegl et al.	10	15–81	8 F 2 M	8 limbs 2 back	various	Benign PEComa	Available in 6 cases: none	not performed
Ueberschaar et al.	1	67	F	wrist	nodule	Benign PEComa	indolent	none mutations
Cohen et al.	1	43	1 M	distal left forearm	nodule	Malignant PEComa	indolent	BIRC3 splice site 1622-27_1631 del37 FANCC R185 * TP53 R248W TSC1 T4151
Kneitz et al.	1	44	/	lower leg	Atypical histiocytoma	Benign PEComa	indolent	not performed

* The real name of mutation.

## Data Availability

Not applicable.

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
