# Peer review of "Primitive Cutaneous (P)erivascular (E)pithelioid (C)ell Tumour (PEComa): A New Case Report of a Rare Cutaneous Tumor"

_genes, 2022, doi:10.3390/genes13071153_

Round 1
Reviewer 1 Report
The manuscript is very well written. I have a few suggestions for the authors:
Abstract, page 1, line 19: Please correct a typo in “Perivascular Epithelioid Cell Tumor (PEComa)”
Abstract, page 1, line 20: Cutaneous primitive PEComa vs primitive cutaneous PEComa. Please make it uniform with the title and through the manuscript. Also, is it “primitive” vs “primary”?
page 1, line 22: cPEComa is an abbreviation but it needs to be fully addressed as “cutaneous PEComa” before its first appearance in the manuscript.
Abstract, page 1, line 25: please provide the full wording of NGS before using it for the first time.
Introduction, page 1, line 33: (P)erivascular (E)pithelioid (C)ells-omas can be modified to PEComa since it’s already appeared in abstract.
Introduction, page 1, line 42: “Primary cutaneous PEComa (cPEComa) are rare…”. Is it “primary” vs “primitive”
In the discussion, the differential diagnosis is brief, other neoplasms need to be discussed including Alveolar Soft Part Sarcoma, Gastrointestinal Stromal Tumor, Granular Cell Tumor, Superficial ALK-rearranged myxoid spindle, Epithelioid Fibrous Histiocytoma, and peripheral nerve sheath tumors
BIRC3 splice site 1622-27_1631 del37, need to be discussed in more detail.
The authors describe the NGS results; it would be better to illustrate the molecular result in a figure to visualize the alignments, with a brief description of the variant calling, including VAF%, etc. Also, The significance of “BIRC3 splice site 1622-27_1631 del37” needs to be classified (as pathogenic, uncertain, or benign). The authors mentioned TFE3 rearrangements/fusions, have they tried FISH or fusion gene analysis?
Lastly, it would be great if the authors briefly summarize the clinicopathologic, molecular, and follow-up information of the previously reported cases in a table.
Author Response
Reviewer n’1: The manuscript is very well written. I have a few suggestions for the authors: Abstract, page 1, line 19: Please correct a typo in “Perivascular Epithelioid Cell Tumor (PEComa)”.
Answer n’1: Dear Reviewer n’1, thank you very much for this kind compliments. We corrected this typo. Sorry. Thanks again.
Reviewer n’1: Abstract, page 1, line 20: Cutaneous primitive PEComa vs primitive cutaneous PEComa. Please make it uniform with the title and through the manuscript. Also, is it “primitive” vs “primary”?
Answer n’2: Done, Thank you very much!
Reviewer n’1: page 1, line 22: cPEComa is an abbreviation but it needs to be fully addressed as “cutaneous PEComa” before its first appearance in the manuscript.
Answer n’3: Dear Reviewer n’1, thank you. We address these question. Thanks again.
Reviewer n’1: Abstract, page 1, line 25: please provide the full wording of NGS before using it for the first time.
Answer n’4: Done, thank you.
Reviewer n’1: Introduction, page 1, line 33: (P)erivascular (E)pithelioid (C)ells-omas can be modified to PEComa since it’s already appeared in abstract.
Answer n’5: Thank you!
Reviewer n’1: Introduction, page 1, line 42: “Primary cutaneous PEComa (cPEComa) are rare…”. Is it “primary” vs “primitive”
Answer n’6: We standardized the terminology.
Reviewer n’1: In the discussion, the differential diagnosis is brief, other neoplasms need to be discussed including Alveolar Soft Part Sarcoma, Gastrointestinal Stromal Tumor, Granular Cell Tumor, Superficial ALK-rearranged myxoid spindle, Epithelioid Fibrous Histiocytoma, and peripheral nerve sheath tumors.
Answer n’7: Dear Reviewer n’1, thank you. We discussed in details all entity that you suggested to us. Thanks again and we hope that paper is more complete, now.
Reviewer n’1: BIRC3 splice site 1622-27_1631 del37, need to be discussed in more detail.
Answer n’8: Dear reviewer, thank you very much for this advice: we explained a little more in detail the meaning of this alteration (aberration) but in Literature it is practically described only once by the recent work of Cohen et al. e, Therefore, we have limited ourselves to observing that we have found it in this case, stressing that in the case of Cohen et al. it was a malignant cPEComa, in our case a benign cPEComa.
Reviewer n’1: The authors describe the NGS results; it would be better to illustrate the molecular result in a figure to visualize the alignments, with a brief description of the variant calling, including VAF%, etc. Also, The significance of “BIRC3 splice site 1622-27_1631 del37” needs to be classified (as pathogenic, uncertain, or benign). The authors mentioned TFE3 rearrangements/fusions, have they tried FISH or fusion gene analysis?
Answer n’9: Dear Reviewer n’1, once again thank you for these important tips. We have a small problem: we do not currently have molecular biology at our institute, which is why we sent the FFPE sample to an affiliated laboratory and whose report was delivered to us in a "descriptive" rather than "figure-based" manner. Therefore, we would feel (if you agree) safer to stay on what we have, as we cannot have a figure that was given to us during molecular analysis. In addition, we examined in even more detail the meaning of the mutation we found, and we implemented it all in the central body of the Manuscript. Let’s hope it works.
Yes, we performed FISH analysis for TFE3 rearrangements with negative results. Thanks again.
Reviewer n’1: Lastly, it would be great if the authors briefly summarize the clinicopathologic, molecular, and follow-up information of the previously reported cases in a table.
Answer n’10: We did. We hope it will work out. Thank you once again.
Reviewer 2 Report
no suggestions
Author Response
Thank you very much. The Authors.